# Mining Natural Product Biosynthesis in Eukaryotic Algae

**DOI:** 10.3390/md18020090

**Published:** 2020-01-30

**Authors:** Ellis O’Neill

**Affiliations:** School of Chemistry, University of Nottingham, University Park, Nottingham NG7 2RD, UK; ellis.oneill@notttingham.ac.uk

**Keywords:** natural products, secondary metabolites, algae, polyketide, non-ribosomal peptide, terpene, genome mining

## Abstract

Eukaryotic algae are an extremely diverse category of photosynthetic organisms and some species produce highly potent bioactive compounds poisonous to humans or other animals, most notably observed during harmful algal blooms. These natural products include some of the most poisonous small molecules known and unique cyclic polyethers. However, the diversity and complexity of algal genomes means that sequencing-based research has lagged behind research into more readily sequenced microbes, such as bacteria and fungi. Applying informatics techniques to the algal genomes that are now available reveals new natural product biosynthetic pathways, with different groups of algae containing different types of pathways. There is some evidence for gene clusters and the biosynthetic logic of polyketides enables some prediction of these final products. For other pathways, it is much more challenging to predict the products and there may be many gene clusters that are not identified with the automated tools. These results suggest that there is a great diversity of biosynthetic capacity for natural products encoded in the genomes of algae and suggest areas for future research focus.

## 1. Introduction

Eukaryotic microalgae are some of the most prolific organisms on the planet, with blooms visible from space [1], and are major contributors to the global carbon cycle. Whilst most algae are benign, harmful algal blooms can have important effects on the environment, causing toxicity and death in marine animals. They also have economic impacts, including in fisheries and the leisure industry. Algae are well-known producers of extremely toxic natural products, including some of the most poisonous small molecules known and cyclic ladder-frame polyethers, compounds not found in bacteria and fungi [2]. These compounds are often synthesised to prevent predation or parasitism of the algal species and are produced under certain environmental conditions or biotic stresses [3].

Algae can broadly be split into the Archaeplastida (Chlorophytes, Rhodophytes and Glaucophytes), whose chloroplast derived from a single endosymbiosis of a cyanobacteria [4], and the secondary algae, which obtained their plastids from a secondary endosymbiosis of another eukaryotic algae [5]. Chlorarachniophytes and Euglenophytes contain chlorophyte derived endosymbionts, independently acquired [6]. Phaeophytes, Haptophytes, Diatoms, Dinoflagellates, Cryptophytes, and Chromerids contain secondary Rhodophyte derived plastids or tertiary plastids derived from one of these other secondary red algae [6]. These complex endosymbioses by free living protists has resulted in integration of genes from the secondary plastid and the nuclear genome of the algal endosymbiont, including genes obtained during primary endosymbiosis, into the host genome. Cryptophytes and Chlorarachniophytes also retain a remnant of the nucleus of the primary endosymbiont, known as a nucleomorph, in close association with the plastids. There is evidence in many of these genomes of lost endosymbionts and the transfer of genes to the nucleus from these cryptic plastids, sometimes referred to as the “shopping bag” model [7].

Researchers are working on sequencing specific algal genomes based on different criteria, from the ease of sequencing and genetic manipulation to their economic and ecological importance [8]. As genome sequencing technology has improved, more algal genomes have become available [9], but these do not capture the full diversity of algae and lag behind sequencing in other classes of organism. Most of the available genomes are from the green algae and many of the others are fragmented due to difficulties in sequencing or assembly of their large and often repetitive genomes. Sequencing of algal genomes reveals that many encode natural product biosynthetic genes similar to those encoded by well-studied bacteria and fungi [10]. Many species harbour some polyketide synthases (PKSs), mostly of the *trans*-acyltransferase type, but there are very few non-ribosomal peptide synthetases (NRPSs) described in algae [11].

In order to understand the biosynthesis of natural products, the genomes of bacteria and fungi are routinely searched for characteristic signatures of biosynthetic gene clusters. Automated tools have been developed, such as antiSMASH (antibiotics & Secondary Metabolite Analysis Shell) [12], which can identify the key biosynthetic genes and related enzymes. This is aided by the tight clustering of genes for the biosynthesis of a single product in gene clusters, particularly in the case of bacterial genomes. Extensive research has also allowed prediction of the substrates used by these enzymes and comparison with characterised biosynthetic gene clusters means it is possible to predict the products of these clusters. In principle, these tools can be used to analyse genomes of other organisms to identify biosynthetic genes. However, organisms outside of the bacterial kingdom often do not have tight enough gene clustering to facilitate automated identification of gene clusters and there may be entirely new classes of biosynthetic genes not identified. Nevertheless, by applying the tools we do have and using careful manual curation, I show herein that it is possible to identify a wide range of natural product biosynthetic genes in eukaryotic microalgae and to make some proposals as to the likely products.

## 2. Results and Discussion

### 2.1. Identifying Natural Product Biosynthetic Genes in Algal Genomes

There are a limited number of genomes available from eukaryotic algae, but as many as possible were selected from different classes of algae and subjected to detailed analysis (see Table 1). It should be noted that algae that have been sequenced have disproportionately small genomes compared to related species, and thus may not truly represent the diversity present within their class. AntiSMASH was used to identify the natural product gene clusters present in these genomes [12]. It was found that using the bacterial version identified gene clusters, but many of these were spurious and the algorithm was not able to efficiently analyse the eukaryotic genome organisation. Occasional identification of arylpolyene or Type II PKSs appeared to be due to fragmentation of genes for fatty acid biosynthesis. It has previously been found that transcriptomes can be analysed using antiSMASH as they contain full length genes with no introns [10]. However, analysing transcriptomes does not give any information about physical clustering of biosynthetic genes in the genome, a hallmark of natural product biosynthesis in other organisms. Using plantiSMASH, designed for identifying the less tightly grouped gene clusters in much larger plant genomes [13], gave no, or only spurious, gene clusters, even in the chlorophytes, which are more closely related to plants. The fungiSMASH algorithm performed best on the algal genomes, although missed some gene clusters found by the bacterial version. It was found that most gene clusters were identified in both algorithms, with a significantly increased identification of terpenes using the bacterial algorithm. For the euglenophytes, only one genome is available, that of *Euglena gracilis* [14], but this is very poorly assembled and indicates only two partial terpene gene clusters, despite previous reports of natural product megasynthases in the transcriptome of this organism [15]. The transcriptome of this species was instead analysed to give a representative distribution for the euglenophytes, though this may not be directly comparable to the genomes used for other species. Manual curation of all of these outputs gave a reliable analysis of natural product biosynthetic genes and gene clusters in these classes of eukaryotic algae (see Figure 1).

Overall the Archaeplastida contained fewer gene clusters than the secondary plastid containing algae, possibly indicating their less complex evolutionary history. All the algae genomes contained some terpene synthases which, on closer inspection, were nearly all related to the enzymes for the biosynthesis of photosynthetic carotenoids. Very few other terpene biosynthetic enzymes were identified. There were a limited number of Type III PKSs (T3PKSs) identified in a wide range of algal genomes, but it is not possible to predict the products of these genes. There were several Type I PKSs (T1PKSs) and a few NRPSs identified, with limited distribution across the algal classes. There was a surprisingly wide distribution of Other(A) genes, which encode an A-domain, a carrier protein and a thioesterase, which are structurally similar to fungal enzymes for the biosynthesis of benzoquinone metabolites [16].

### 2.2. Distribution of Biosynthetic Gene Clusters Amongst Algal Classes

#### 2.2.1. Green Algae (Chlorophytes)

Green algae are the most highly studied group of algae and have a wide range of genomes available. Analysing the genomes of 22 different species using antiSMASH and fungiSMASH revealed an average of 5.3 gene clusters per species. Terpenes were the most abundant type of gene cluster found, with an average of 2.5 gene clusters per species, likely involved in carotenoid biosynthesis. T3PKS gene clusters were also common, with an average of 1.5 T3PKS genes per species.

In many of the genomes analysed, there is one T1PKS, which is annotated by antiSMASH as NRPS-like. This is often present as one megasynthase with an adenylation domain at the N-terminus and 10–11 highly reducing PKS modules (see Figure 2). In some species this megasynthase is split across different polypeptides, and these are sometimes not found on the same contig, but close inspection indicates they are all have close homology to the single megasynthase of *Chlamydomonas reinhardtii*. It is unclear whether this fragmentation is due to sequencing or assembly errors, or whether the megasynthase is spread across the genome in these organisms. It has not been possible to isolate the product formed but some aspects of its structure can be predicted [17]. The megasynthases do not contain any acyl transferase (AT) domains, which is presumed to be encoded as another gene, and thus are designated *trans*-AT PKSs. The starter unit, activated by the A-domain, cannot be predicted with the current tools. The lack of any enzymatic domain responsible for reduction in the two final modules suggest this megasynthase may produce a triketide-like molecule which could spontaneously cyclise to form a hydroxy pyrone (see Figure 2). Disrupting this gene in *Chlamydomonas* prevented the correct development of zygotes and so the product of this PKS is proposed to be a structural component of the zygote cell wall [17].

#### 2.2.2. Red Algae (Rhodophytes)

The available genomes from the red algae are mostly from multicellular seaweeds, but there are two single celled species represented. The genomes do not contain many gene clusters but do have several terpene biosynthetic genes and a single T3PKS. *Kappaphycus alvarezii* has a substantially larger genome than the other species and also encodes a T1PKS with other biosynthetic enzymes in close proximity. The PKS contains a phosphopantetheinyl transferase, for activating the acyl carrier proteins, and an acetyl-CoA carboxylase, for the synthesis of the extension units of the PKS. The other domains are not in the typical order found in PKSs and this megasynthase may possibly be involved in the synthesis of polyunsaturated fatty acids. A partial T1PKS can be identified in the *Gracilariopsis lemaneiformis* genome, but the fragmented nature of the genome leaves it unclear if this is a reliable annotation.

#### 2.2.3. Glaucophytes

*Cyanophora paradoxa* is the only genome sequenced member of the Glaucophytes, a small group of freshwater primary endosymbionts. It encodes two terpene synthases and one NRPS. This contains two condensation (C) domains and, although some of the acyl carrier proteins (ACPs) are not very well predicted, likely makes a tripeptide. The amino acids incorporated by this NRPS cannot be predicted with the current tools.

#### 2.2.4. Brown Algae (Phaeophytes)

Of the brown algal genomes available for analysis, all three are from macroalgae. Brown algae are well known producers of phlorotannins, bioactive polyphenols with antibacterial and antioxidant properties, which are synthesised by T3PKSs [18]. The genome of *Saccharina japonica* contained just one T3PKS, a terpene synthase and an Other(A) protein, though on a small contig fragment. *Ectocarpus siliculosus* contained a single Other(A) gene, a terpene synthase and three T3PKSs. The genome of *Cladosiphon okamuranus* on the other hand has one highly fragmented NRPS and three T1PKSs. Two of these have single fully reducing PKS modules and terminating amino transferase domains and the other T1PKS cluster contains two KS domain containing proteins, similar to those for the biosynthesis of heterocyst glycolipids from the cyanobacteria *Nostoc punctiforme* [19]. The presence of these two adjacent genes and proximity to other putative biosynthetic genes, may indicate some gene clustering is present in this organism. In addition, there were ten terpene synthase genes and two T3PKSs encoded in the *C. okamuranus* genome.

#### 2.2.5. Haptophytes

Haptophytes are a class of secondary endosymbiotic algae, well known for forming very large blooms and producing complex toxins. Several PKSs have previously been identified in the genome of *Emiliania huxleyi* [20], and in expressed sequence tags (ESTs) from the toxin producing species *Chrysochromulina polylepis* [21], but it was unclear whether these were more universally distributed among the haptophytes. With two genomes of *Chrysochromulina* (*tobinii* and *parva*) also now available, it is clear that these algae contain an abundance of PKSs. It is possible to link more closely related PKSs based on the sequence of their ketosynthase (KS) domains (for example see Figure 3), as those that are involved in making a single product have a higher sequence homology [22].

In these analysed haptophyte genomes, there are an average of six Other(A) genes per species, suggesting that the Haptophytes are prolific producers of these metabolites. They also encode three T3PKSs and three terpene synthases in the genomes. All three species contain two unusual T1PKSs with their KS domains more closely related to each other than to any of the other KS domains in the genomes (see Figure 3), with both proteins containing an A-domain (see Figure 4a). One of these is predicted to be specific for alanine and encodes a PKS module which forms an alcohol and two thioesterases, though it is not clear how both would be involved in the biosynthesis of the product. The second of these T1PKS has a fully reducing PKS module with the ketoreductase (KR) out of the usual order for T1PKSs. It is unclear if these are both involved in the biosynthesis of the same compound.

Both *Chrysochromulina* genomes encode a hybrid NRPS-PKS (Figure 4b), which contains several unusual features, including two amino transferases and a methyl transferase [23]. Careful analysis of the domain structure allows the product to be partially predicted [10]. Additionally, in *C. parva* there is an apparent PKS-NRPS hybrid, which contains an *N*-acetyl transferase, a KS domain, a C domain and an adenylation (A) domain, predicted to be specific for cysteine (see Figure 4c). As this is located on a small contig and the KS domain is no more related to other KS domains in this species, it is not possible to predict the whole protein structure or any related enzymes and thus it is not possible to predict the final product this module contributes to synthesising.

*E. huxleyi* encodes eight T1PKSs, including one that appears to be involved in the synthesis of poly-unsaturated fatty acids. There are two PKSs which encode AT, KS and ACP modules, one with an additional KR with a termination domain, and one with a predicted methyl transferase and a dehydrogenase (DH) domain. Two proteins that are in close proximity on the genome encode PKS domains in an extremely unusual order (see Figure 4d). All of the modules required for a fully reducing PKS module are present, as well as an extra KS domain and an amino transferase. With a reductive offloading, a tentative structure prediction can be proposed. The next smallest PKS in the *Emiliana* genome contains three fully reducing modules, a KR containing module and a sulfotransferase (see Figure 4e). Two more large, multi-modular PKSs are present in the *E. huxleyi* genome, both with three modules containing two enoyl hydratase and hydroxymethylglutaryl-CoA synthase, which are proposed to add a beta methyl group, as in the biosynthesis of bacillaene [24]. The rest of these megasynthases seem to follow the standard PKS biosynthetic logic and so products can be proposed with some reasonable confidence (see Figure 4f,g).

Notably all the PKSs from these Haptophytes contain a single AT domain at the N-terminus of the megasynthases. There are two well characterised sub types of T1PKS: *cis*-AT, with a unique AT domain in every module, each able to activate a different substrate; and *trans*-AT with the AT domain as a separate protein, acting for all modules. With a single AT domain in the megasynthase, these haptophytes appear to have a “semi-*trans*”-AT domain, which activates all the precursors for the KS domains in each protein. Each extension will therefore use the same precursor which is like to be the most commonly used PKS extender unit, malonyl-CoA.

#### 2.2.6. Diatoms (Bacillariophyta)

Three Diatom genomes are available, with approximately 11,000 genes on their ~30 Mbp genomes [25]. The three Diatoms only have approximately seven gene clusters—over half of which are related to terpene biosynthesis, probably for the production of photosynthetic carotenoids. Interestingly, all three genomes contain an Other(A), with an A domain, a ACP domain, a reductive termination enzyme and a KR. It is unclear what the product of this enzyme might be. *Phaeodactylum tricornutum* also contains three more Other(A) proteins and *Thalassiosira pseudonana* has a protein with both A and C domains, though this is on a small fragmented contig and it is unlikely to be full length.

#### 2.2.7. Dinoflagellates

Dinoflagellates are well known to produce a wide array of extremely large and structurally complex compounds—many of which are harmful to human health [26]. They are found as free living algae, but also form endosymbiotic relationships with coral and sponges and their genome size varies hugely from 0.5–185 Gbp [27]. Some investigation into the biosynthesis of toxins has been undertaken in Dinoflagellates, based on ESTs or transcriptomic approaches. For example, the ~35 Gbp genomes of *Gambierdiscus* species were found to contain approximately 100 KS domains—more than enough for the biosynthesis of the enormous 164 carbon containing polycyclic polyether maitotoxin [28]. A close homologue of the key cyanobacterial PKS for the biosynthesis of the neurotoxin saxitoxin, *sxtA*, was identified in the genome of the saxitoxin producing Dinoflagellate *Alexandrium* [29].

Few Dinoflagellate genomes are available in the NCBI database and they are all exceptionally small, incomplete and highly fragmented. Although these are not representative of this group of algae, some gene clusters can be identified using antiSMASH. All analysed genomes encode a terpene synthase and at least one Other(A) protein containing an A-domain and a carrier protein, often with one or two extra domains. The genome of *Prorocentrum minimum* was highly fragmented but contained partial sequences for multidomain PKS and NRPS megasynthases. The genome of *Breviolum minutum* encoded a hybrid PKS-NRPS with several unusual features including the lack of some expected carrier proteins, an A domain with no associated C domain and a oxidoreductase domain with no clear substrate but which may act upon the side chain of an amino acid (see Figure 5). This gene also encodes a semi-*trans*-AT domain. These unusual features make predicting the structure of the product highly uncertain.

#### 2.2.8. Cryptophytes

The only Cryptophyte whose genome has been sequenced is *Guillardia theta*, which has a secondary endosymbiont that retains a nucleomorph remnant of the ancestral endosymbiont’s nucleus. It contains eight terpene synthases, all predicted to be involved in the biosynthesis of photosynthetic pigments. There were also five Other(A) genes with an A-domain, a carrier protein and another domain, such as thioesterases and thioester reductases.

#### 2.2.9. Chromerids

Chromerids are single celled photosynthetic algae closely related to the apicomplexan parasites, such as malaria. They contain a chloroplast of red algal origin, thought to have been obtained by tertiary endosymbiosis of a heterokont alga. There are two genomes available from these algae, that of *Chromera velia* and *Vitrella brassicaformis* [30]. Both genomes contain many T1PKS with *cis*-AT domains and almost every module is fully reducing, likely forming long alkyl chains. There are also several terpene synthases, a very small fragment of an NRPS and 2–3 Other(A) genes in each genome.

#### 2.2.10. Chlorarachniophytes

The genome of *Bigelloweilla natans* is the only genome available from the Chlorarachniophytes. It has previously been reported that the *B. natans* genome encodes a three domain NRPS [11]. In this analysis, several NRPS genes and one PKS were identified in the genome (see Figure 6). The three domain NRPS appears to contain an epimerase and a very tentative assignment for selectivity towards cysteine. There is also a two-domain NRPS with a reductive terminating enzyme, which may cyclise the product to form a keto piperazine. The genome encodes a singly reducing PKS with a *cis*-AT domain and a single module of an NRPS with a reductive thioesterase. It is possible that these latter two act together to make a single product, based on the compatibility of the C- and N-terminal domains, though there is no direct evidence for this.

#### 2.2.11. Euglenophytes

Transcriptome sequencing of *Euglena gracilis* unveiled a wide range of natural product biosynthetic enzymes, including PKSs and NRPSs [31]. Surprisingly, these were not identified at all using antiSMASH on the recently sequenced genome [14]. It should be noted that this genome is highly fragmented and is the largest algal genome sequenced to date. Interrogating the transcriptome using antiSMASH revealed many natural product gene clusters, including three T1PKSs and six NRPSs (see Figure 7). These may not be full length due to the limitations of transcriptome sequencing. It is unclear which of these proteins might act together in the synthesis of one molecule. Notably the full T1PKS domains all have *cis*-AT domains. There were also seven Other(A) type enzymes identified, some with reducing terminating domains and some with hydrolysing thioesterases.

## 3. Conclusions

Algal genomes encode a wide range of natural product biosynthetic enzymes and different classes of algae contain different categories, such as the larger number of T1PKSs in the Chromerids and the presence of several NRPSs in the Euglenophytes. Some algal classes, such as the Haptophytes, have many more diverse biosynthetic genes, whilst others, such as the Chlorophytes and Rhodophytes, are much more restricted. This may be related to the complex evolutionary history of the algal classes that have a secondary plastid. It is notable that there are some class specific gene clusters, such as the 11 module *trans*-AT T1PKS in the Chlorophytes and the conserved Other(A) in the Diatoms. Although there are not many genera represented more than once, both *Chrysochromulina* species contain matching hybrid NRPS-PKS genes.

There are many good examples of multi module NRPS and T1PKS megasynthases present in these algal genomes. It is remarkable that in these incredibly diverse organisms the domain architecture of the megaynthases is conserved between the algae, bacteria and fungi, suggesting either an ancient evolutionary origin or a high degree of horizontal gene transfer between these different organisms. Interestingly, all the T1PKSs from the Haptophytes and the Dinoflagellate *Breviolum minutum* contain an N-terminal semi-*trans*-AT domain, not noted in other organisms. Due to the phylogenetic distance of algae from each other and the well-studied bacteria and fungi, the predictions of amino acids or ketide units used by NRPSs and PKSs are not reliable, making it difficult to predict the structure of the product synthesised by these genes/gene clusters. There are notably no ribosomally synthesized and post-translationally modified peptides (RiPPs) discovered in any of the genomes analysed. These are annotated based on the presence of tailoring enzymes, which may not be accurately identified in the algal genomes.

Algae which have had their genomes sequenced to date have relatively small genomes, limiting this study. The quality of the genomes varies significantly, meaning that it is very difficult to identify gene clusters that are similar in structure to the dense gene clusters found in bacteria and fungi. There are some examples of clustering of biosynthetic genes apparent, but these do not contain many genes and it is unclear if they encode all the proteins necessary for the synthesis of a single product, or whether tailoring genes are found elsewhere on the genome. If more, better quality, algal genomes become available, the nature of the natural product biosynthetic genes and any clustering in algae can be further explored. The biological role of the products of these gene clusters is not yet clear, and their structures can only be tentatively predicted using bioinformatics. Some may act as structural components, such as the Chlorophyte polyketide [17], some as antifeedants, such as the Euglena ichthyotoxins [32], and some may be as yet undiscovered antibiotics. Antibacterial and antifungal compounds are likely to be highly valuable to algae, as they need to protect themselves against microbial pathogens in the environment. Synthesis of antibacterials would be particularly interesting, as the algae would not need resistance determinants to prevent self-toxicity, unlike in bacteria [33].

These results indicate that algae contain a wide range of natural product biosynthetic genes, though studies into these and their corresponding natural products are limited. Future sequencing of more algal genomes will shed light on the biosynthetic capability of algae, particularly in those classes with secondary plastids, as their genomes seem to harbour more biosynthetic gene clusters. Further work will be needed in the under surveyed classes, such as the Cryptophytes, Chlorarachniophytes and Euglenophytes, to explore the full potential encoded by theses algae. There are limitations with the current prediction tools available for algal genomes and these can be addressed by refining these algorithms. The available tools perform better on the transcriptomic data than on genomic data, though this precludes analysis of gene clustering, and that the diversity of natural product genes is likely to be underestimated in this analysis. Isolation and structural elucidation of more bioactive compounds from algae and the identification of their corresponding gene clusters may ultimately help to inform the development of reliable prediction tools.

Even within the limited range of genomes available, the wide range of natural product biosynthetic genes, with both familiar and novel features, show that algae are a promising source of new biosynthetic pathways and novel natural products, ripe for exploitation.

## 4. Materials and Methods

### Gene Cluster Identification

In order to identify natural product biosynthetic gene clusters, assembled genomes from all available eukaryotic microalgae, with 22 representative chlorophytes, were downloaded from the NCBI. These were analysed using antiSMASH, plantiSMASH and fungiSMASH using the standard parameters [12]. If necessary, very large files were split arbitrarily into smaller files for upload. Gene clusters identified were subject to manual curation and ectoine, siderophore, bacteriocin and homoserine lactone gene clusters were not considered further in this study. NRPSs were identified as such if they contained both condensation (C) and adenylation (A) domains. Other(A) are a class of enzymes that contain an A-domain and a carrier protein with some other domain such as a thioesterase, sometimes identified by antiSMASH as NRPSs or NRPS-like, and are related to proteins for the synthesis of fungal benzoquinone metabolites [16]. T1PKSs are annotated as such if they contain multiple domains including at least one KS domain and T3PKSs are annotated as identified by antiSMASH.

Proteins identified as being involved in natural product biosynthesis were subject to further analysis using BLAST and CDD search, with larger regions of the genome (~20 kb) searched if it was suspected that proteins had been incorrectly annotated or truncated [34]. This was particularly problematic when using the bacterial algorithm, where there was incorrect intron identification, and for genes close to the end of contigs being incomplete. Proteins identified as having a KS domain were subject to further analysis using NaPDoS to identify more closely related genes [22]. KS domains were compared within and between species in the same algal class. Products were predicted based on sequential action of the domains present in the megasynthases, assuming malonate as the extender unit and with amino acids only annotated if predicted by antiSMASH.

## Figures and Tables

**Figure 1 marinedrugs-18-00090-f001:**
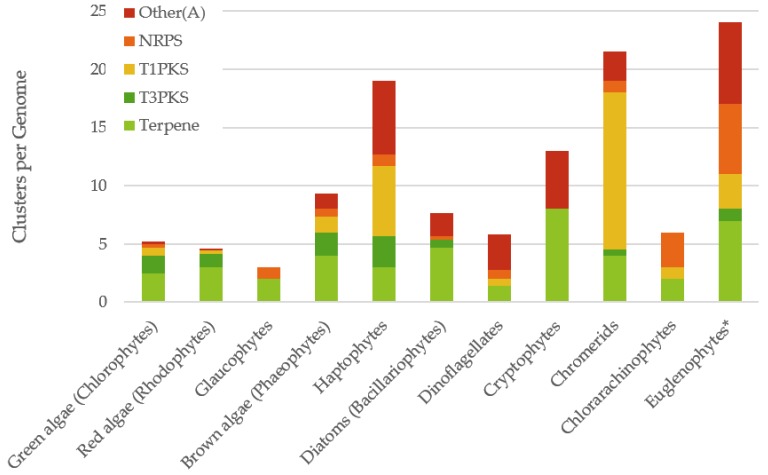
Distribution of gene cluster types in algal genomes. * Note the Euglenophyte data is based on the transcriptome of *Euglena gracilis*. NRPS = Non-ribosomal peptide synthetases, T1PKS = Type I polyketide synthases, T3PKS = Type III polyketide synthases.

**Figure 2 marinedrugs-18-00090-f002:**
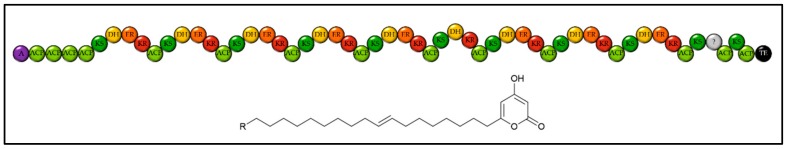
Domain structure of the conserved large *trans*-AT PKS from *Chlamydomonas reinhardtii*. This gene is encoded at locus CHLRE_10g449750v5. A = Adenylation domain. ACP = Acyl carrier protein. KS = Ketosynthase. DH = Dehydratase. ER = Enoyl Reductase. KR = Ketoreductase. TE = Thioesterase. ? denotes a probable discreet domain for which no function can be predicted and which may not be functional.

**Figure 3 marinedrugs-18-00090-f003:**
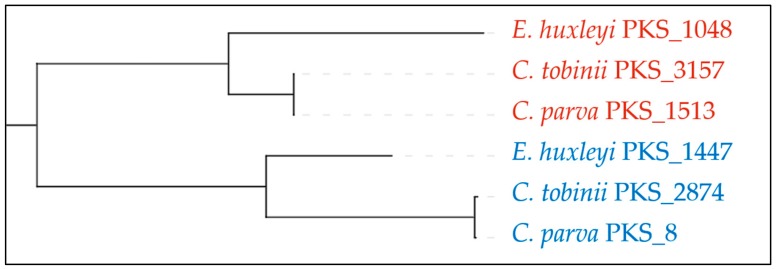
Phylogeny of selected KS domains for selected PKSs in haptophytes generated using Natural Product Domain Seeker (NaPDoS) [22]. Three ketosynthase (KS) domains, one from each species, are more closely related to each other (red), indicating these are likely to be the same module performing the same reaction on one molecule. These are more closely related to another set of KS domains (blue) than any others (not shown) in the genome of these organisms and are more likely to act on the same molecule, as part of a different PKS module. Numbers indicate the contig these KS domains are located on.

**Figure 4 marinedrugs-18-00090-f004:**
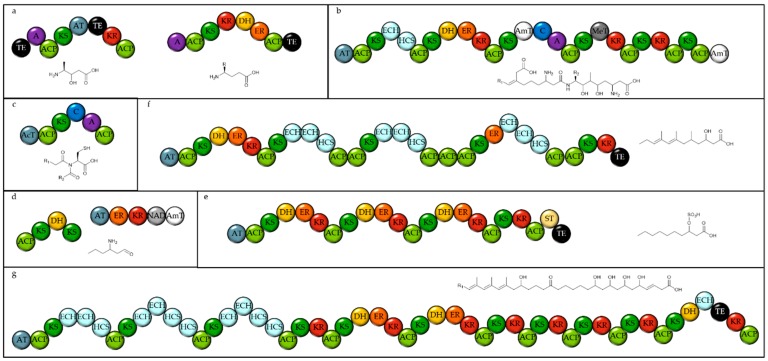
Domain structure of the natural product megasynthases from Haptophytes and tentatively proposed structures for their products. (**a**) The PKSs shared amongst all three Haptophytes. (NCBI accession KB864057, PJAB01001513 and JWZX01003157 and KB864456, PJAB01000008 and JWZX01002874 in *Emiliania huxleyi*, *Chrysochromulina parva* and *C. tobinii* respectively) Both have unusual domain architectures. (**b**,**c**) The PKSs found in both *Chrysochromulina* species ((**b**) PJAB01000005 and JWZX01003258, (**c**) PJAB01001579.1 and JWZX01001313 in *C. parva* and *C. tobinii* respectively). (**d**–**g**). PKSs only found in *E. huxleyi* (KB864737, KB864225, KB867561 and KB868012). The substrate specificity of the A domains is not clear. TE = Thioesterase. A = Adenylation domain. ACP = Acyl carrier protein. KS = Ketosynthase. AT = Acyl transferase. KR = Ketoreductase. ECH = Enoyl-CoA-hydratase. HCS = Hydroxymethylglutaryl-CoA synthase. DH = Dehydratase. ER = Enoyl Reductase. AmT = Amino transferase. C = Condensation domain. MeT = Methyl transferase AcT = N-Acetyl Transferase. NAD = NAD dependent oxidoreductase. ST = Sulfotransferase.

**Figure 5 marinedrugs-18-00090-f005:**
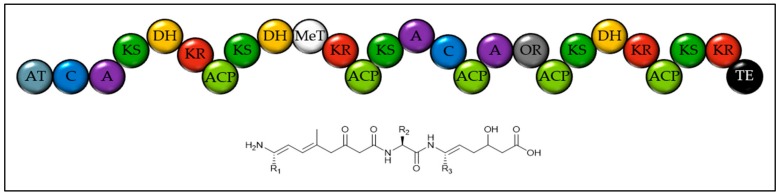
Domain structure of the large hybrid NRPS-PKS from *Breviolum minutum* and tentatively proposed structure of the product. This gene is found on contig 109 (NCBI accession DF239860) and needed extensive manual curation to identify all domains. The amino acid specificity of the A domains is not clear. AT = Acyl transferase. ACP = Acyl carrier protein. C = Condensation domain. A = Adenylation domain. KS = Ketosynthase. DH = Dehydratase. KR = Ketoreductase. AmT = Amino transferase. MeT = Methyl transferase. OR = Oxidoreductase. TE = Thioesterase.

**Figure 6 marinedrugs-18-00090-f006:**
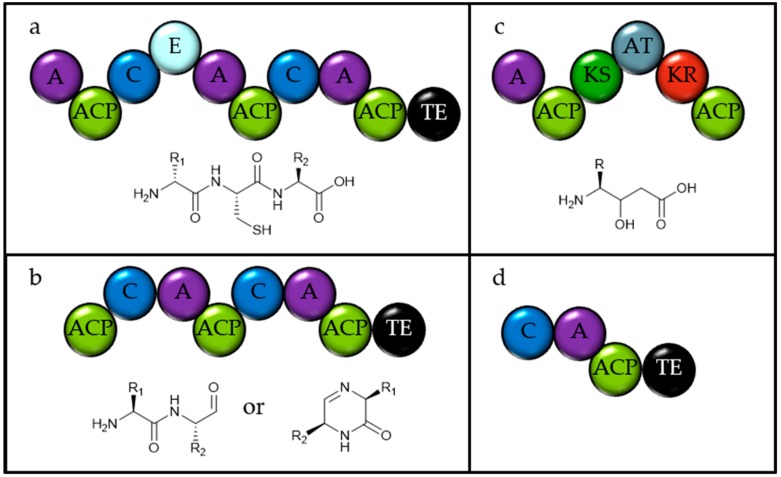
Domain structure of the natural product megasynthases from *Bigelloweilla natans* and tentatively proposed structure of the products. (**a**) Gene 18 on contig 2069 (NCBI accession ADNK01001542.1) (**b**) Gene 5 on contig 690 (NCBI accession ADNK01003012.1). (**c**) Gene 25 on contig 859 (NCBI accession ADNK01002829.1). (**d**) Gene 6–7 on contig 3305 (NCBI accession ADNK01000237.1). A = Adenylation domain. ACP = Acyl carrier protein. C = Condensation domain. E = Epimerase. KS = Ketosynthase. AT = Acyl transferase. KR = Ketoreductase. TE = Thioesterase.

**Figure 7 marinedrugs-18-00090-f007:**
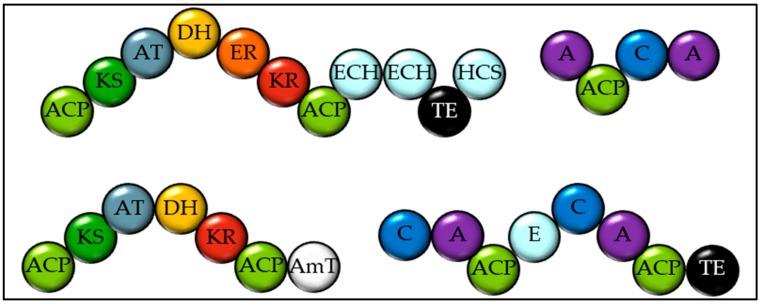
Domain structure of the four largest megasynthases from *Euglena gracilis*. EBI accession numbers GEFR01000028, GEFR01000292, GEFR01000082 and GEFR01000053. ACP = Acyl carrier protein. KS = Ketosynthase. AT = Acyl transferase. DH = Dehydratase. ER = Enoyl Reductase. KR = Ketoreductase. ECH = Enoyl-CoA-hydratase. HCS = Hydroxymethylglutaryl-CoA synthase. A = Adenylation domain. C = Condensation domainTE = Thioesterase. AmT = Amino transferase.

**Table 1 marinedrugs-18-00090-t001:** Gene clusters identified in algal genomes.

Class	Species	Ungapped Sequence Length	Accession	Number of Gene Clusters				
Green Algae (Chlorophytes)				Terpene	T3PKS	T1PKS	NRPS	Other(A)
	*Chlamydomonas reinhardtii*	107,048,224	GCA_000002595	1	3	1	0	0
	*Micromonas pusilla*	21,706,984	GCA_000151265	5	0	2	0	0
	*Volvox carteri f. nagariensis*	125,467,762	GCA_000143455	3	2	0	0	0
	*Chlorella variabilis*	42,214,557	GCA_000147415	5	1	0	1	0
	*Coccomyxa subellipsoidea C*-*169*	48,826,616	GCA_000258705	1	5	1	0	0
	*Auxenochlorella pyrenoidosa*	48,566,231	GCA_001430745	2	0	0	0	0
	*Helicosporidium sp.*	12,373,820	GCA_000690575	1	2	0	0	0
	*Parachlorella kessleri*	59,187,803	GCA_001598975	1	0	1	2	0
	*Prototheca cutis*	19,644,471	GCA_002897115	2	1	1	0	0
	*Eudorina sp.*	182,993,185	GCA_003117195	2	3	0	0	0
	*Yamagishiella unicocca*	134,234,618	GCA_003116995	1	1	0	0	1
	*Trebouxia gelatinosa*	60,898,934	GCA_000818905	1	0	0	0	0
	*Micractinium conductrix*	61,018,900	GCA_002245815	2	0	0	0	0
	*Dunaliella salina*	280,838,039	GCA_002284615	2	1	0	0	0
	*Botryococcus braunii*	179,769,887	GCA_002005505	2	4	0	0	0
	*Tetrabaena socialis*	97,974,014	GCA_002891735	2	0	0	0	0
	*Picocystis sp. ML*	29,646,247	GCA_003665715	7	3	3	0	0
	*Ostreococcus tauri*	14,758,467	GCA_002158475	5	0	1	0	2
	*Gonium pectorale*	117,596,311	GCA_001584585	2	3	1	0	0
	*Cymbomonas tetramitiformis*	262,008,979	GCA_001247695	1	3	4	3	2
	*Klebsormidium nitens*	103,146,182	GCA_000708835.1	2	0	0	0	1
	*Chara braunii*	1,429,941,810	GCA_003427395	5	1	0	0	0
	**Average**			**2.5**	**1.5**	**0.7**	**0.3**	**0.3**
Red Algae (Rhodophytes)								
	*Gracilariopsis chorda*	92,180,038	GCA_003194525	3	0	0	0	0
	*Porphyridium purpureum*	19,451,899	GCA_000397085	1	1	0	0	0
	*Galdieria sulphuraria*	13,419,354	GCA_000341285	3	1	0	0	0
	*Chondrus crispus*	104,085,276	GCA_000350225	3	1	0	0	0
	*Porphyra umbilicalis*	87,766,581	GCA_002049455	4	1	0	0	0
	*Gracilariopsis lemaneiformis*	86,759,375	GCA_003346895	4	1	1	0	1
	*Kappaphycus alvarezii*	336,721,358	GCA_002205965	3	3	1	0	0
	**Average**			**3**	**1.1**	**0.3**	**0**	**0.1**
Glaucophytes								
	*Cyanophora paradoxa*	99,940,401	GCA_004431415	2	0	0	1	0
Brown Algae (Phaeophytes)								
	*Ectocarpus siliculosus*	191,106,465	GCA_000310025	1	3	1	0	1
	*Saccharina japonica*	537,522,535	GCA_000978595	1	1	0	0	1
	*Cladosiphon okamuranus*	166,898,169	GCA_001742925	10	2	3	1	3
	**Average**			**4**	**2**	**1.3**	**0.3**	**1.7**
Haptophytes								
	*Emiliania huxleyi*	155,930,723	GCA_000372725	2	2	8	0	8
	*Chrysochromulina parva*	65,764,750	GCA_002887195	3	3	5	2	5
	*Chrysochromulina tobinii*	59,073,094	GCA_001275005	4	3	5	1	6
	**Average**			**3**	**2.7**	**6**	**1**	**6.3**
Diatoms (Bacillariophytes)								
	*Thalassiosira pseudonana*	32,272,629	GCA_000149405	5	1	0	0	1
	*Thalassiosira oceanica*	92,185,637	GCA_000296195	4	0	0	1	1
	*Phaeodactylum tricornutum*	27,017,695	GCA_000150955	5	1	0	0	4
	**Average**			**4.7**	**0. 7**	**0**	**0.3**	**2**
Dinoflagellates								
	*Symbiodinium microadriaticum*	745,992,902	GCA_001939145	1	0	2	1	5
	*Symbiodinium sp. clade A Y106*	756,831,958	GCA_003297005	2	0	0	0	1
	*Symbiodinium sp. clade C Y103*	674,313,450	CA_003297045	1	0	0	0	3
	*Breviolum minutum*	603,733,232	GCA_000507305	1	0	1	2	5
	*Prorocentrum minimum*	29,349,011	GCA_001652855	2	0	0	1	1
	**Average**			**1.4**	**0**	**0.6**	**0.8**	**3**
Cryptophyte								
	*Guillardia theta*	83,457,412	GCA_000315625	8	0	0	0	3
Chromerids								
	*Vitrella brassicaformis*	71,768,979	GCA_001179505	3	0	18	1	2
	*Chromera velia*	187,454,854	GCA_000585135	5	1	9	1	3
	**Average**			**4**	**0.5**	**13.5**	**1**	**2.5**
Chlorarachinophytes								
	*Bigelowiella natans*	91,405,885	GCA_000320545	2	0	1	3	0

Euglenophytes								
	*Euglena* gracilis	1,435,499,417	GCA_900893395	7	1	3	6	7

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
