# Peer review of "Mining Natural Product Biosynthesis in Eukaryotic Algae"

_marinedrugs, 2020, doi:10.3390/md18020090_

Round 1

Reviewer 1 Report

Basically, the idea to address "algae" in this context, i. e., their biosynthetic potential for producing small natural products, is worthwhile to be presented and discussed in a broader context.

There are two major issues that let me suggest a minor revision of the manuscript.

1) Results and discussion seems to be mixed in several places. I regard the genome source (Supplementary table) as essential for the reader and recommend to have it as a part of the manuscript (can be added and described in Materials and Methods). Since data sets are used that have to be referred to in all places of the manuscript the separation of results and discussion should be avoided. Instead a 'Conclusion' may be added.

2) The introduction is not sufficient as far as the evolution and taxonomy of 'algae' is concerned. A recent monograph concerning algae and all the terminology used therein should be used. For example 'diatoms' is science slang for 'Bacillariophyta". An evolutionary tree would be helpful to 'localize' the group of organisms which is addressed by the results. Please also use the correct taxonomical terms. It is difficult within the 'algae' but if the appropriate reference is given this issue can be solved easily.

3) Also the endosymbiosis issue should be considered more seriously. Which photosynthetic machinery was obtained from a cyanobaterium (primary), which one from a 'red algae'. This can also be used for organizing the manuscript in a more 'systematic' way. What about the Charophyta? Wrong spelling in Chlorarachniophyta. (see and cite , e.g., ï¿¼Miroslav Oborníkc, Endosymbiotic Evolution of Algae, Secondary Heterotrophy and Parasitism, Biomolecules 2019, 9(7), 266; https://doi.org/10.3390/biom9070266

Author Response

I thank the reviewer for their comments which address in the order they have been suggested.

1) The results and discussion have been combined and the conclusions placed separately. The supplementary table has been added to the materials and methods. as suggested.

2)The evolution and taxonomy has been expanded in the introduction and further references added. I have not included an phylogenetic tree as many lineages are difficult to root correctly and their relationships are still subject to much controversy. 

3)The endosymbiosis is considered more thoroughly in the introduction, the citation added as suggested and the spelling corrected. Two Charophytes and two Chromerids have also been added. The whole manuscript has been reordered in the systematic way recommended. 

Reviewer 2 Report

In the manuscript by O’Neill, the author explores the distribution and diversity of natural product genes among available and representative algal genomes. More specifically, the author uses the antiSMASH, plantiSMASH, and fungiSMASH pipelines along with manual curation to identify polyketide, non-ribosomal peptide, and terpene biosynthetic gene clusters in select Archaeplastida and secondary plastid containing algae. It was found that all of the organisms investigated contained multiple natural product gene clusters and that in general the Archaeplastida contained fewer gene clusters than the other species investigated. The domain structures and predicted products from several identified natural product megasynthases were also reported. This manuscript will be of potential interest to the readership of Marine Drugs, provided the following comments/concerns are addressed.

It was noted in the Results section that the various SMASH algorithms used often gave no or spurious results for natural product gene cluster identification, and that for one species, Euglena gracilis, transcriptome data needed to be used to obtain any results at all (and this species was found to contain the most natural product gene clusters of any examined). Furthermore, it seems manual curation was only performed on the outputs of these analyses. Therefore, it is unclear how robust the analysis was and the extent to which gene clusters were missed, which brings into question the meaningfulness of the relative gene cluster distribution data among the algae (Figure 1). It was stated that better bioinformatic tools are needed for cluster identification in algae, and this is one conclusion that can be drawn. Another may be that the available tools perform better on the transcriptomic data than on genomic data (likely due to the absence of introns), and that the diversity of natural product genes is likely to be underestimated in this analysis. At any rate, these points should be addressed and clarified in the text, and conclusions drawn about the relative abundance of natural product gene clusters in the various types of algae (which is a major point of the article) may need to be tempered. No comments were made as to whether any of the identified clusters could be responsible for/assigned to the biosynthesis of natural products that are known to be produced by a given species. This would be a very valuable/insightful addition to the manuscript and provide further confirmation that the analysis hasn’t missed several important natural product gene clusters. The description of KS domain homology in the Haptophyte section and Figure 3 is somewhat confusing and could stand to be rephrased. It was stated several times in the text that it was ‘impossible’ to predict the products of some of the identified gene clusters. These sentences should be rephrased to something less absolute (e.g., difficult using conventional methods, attempts failed, etc).

Author Response

I thank the reviewer for their comments, which I have responded to below. 

The use of antiSMASH to interrogate the genome allows the identification of multiple biosynthetic genes on the same region of the genome, which often act in the biosynthesis of a single natural product in other organisms. This information is lost in the transcriptome, and so the method used here was designed to retain this information. The manual curation was performed to ensure all information of identified clusters was extracted and not lost, but the SMASH algorithms were used to identify the core biosynthetic enzymes. I agree that this may miss some clusters, but by applying the same protocol to all the genomes the results should be comparable. A larger concern, which I raise in the manuscript, is that the sequenced genomes are not representative of their phyla, typically being smaller and less complex to aid sequencing. I hope this will stimulate the community to look at a wider diversity of algae.

None of these gene clusters have been linked directly to a product. None of these sequenced strains have been thoroughly explored for the production of natural products as yet and so no link can be made.

The description regarding the KS domains in the Haptophyte section and figure 3 have been clarified. The references to "impossible" have been replaced as suggested.